# Evaluation of NPP using three models compared with MODIS-NPP data over China

**Jinke Sun**[1], **Ying Yue**[2], **Haipeng Niu**[1] *

**1** School of Surveying and Land Information Engineering, Henan Polytechnic University, Jiaozuo, Henan, China, **2** School of Emergency Management, Henan Polytechnic University, Jiaozuo, Henan, China

* niuhaipeng@126.com

**Data Availability Statement:** Regarding data availability: 1. https://lpdaac.usgs.gov/data_access/ This is the website where you can download the data: MOD17A3 data https://lpdaac.usgs.gov/products/mcd12q1v006/#tools MCD12Q1 data

## Abstract

Estimating net primary productivity (NPP) is significant in global climate change research and carbon cycle. However, there are many uncertainties in different NPP modeling results and the process of NPP is challenging to model on the absence of data. In this study, we used meteorological data as input to simulate vegetation NPP through climate-based model, synthetic model and CASA model. Then, the results from three models were compared with MODIS NPP and observed data over China from 2000 to 2015. The statistics evaluation metrics (Relative Bias (RB), Pearson linear Correlation Coefficient (CC), Root Mean Square Error (RMSE), and Nash-Sutcliffe efficiency coefficient (NSE)) between simulated NPP and MODIS NPP were calculated. The results implied that the CASA-model performed better than the other two models in terms of RB, RMSE, NSE and CC whether on the national or the regional scale. It has a higher CC with 0.51 and a smaller RMSE with 111.96 g C·m⁻²·yr⁻¹ in the whole country. The synthetic model and CASA-model has the same advantages at some regions, and there are lower RMSE in Southern China (86.35 g C·m⁻²·yr⁻¹), Xinjiang (85.53 g C·m⁻²·yr⁻¹) and Qinghai-Tibet Plateau (93.22 g C·m⁻²·yr⁻¹). The climate-based model has widespread overestimation and large systematic errors, along with worse performances (NSEmax = 0.45) and other metric indexes unsatisfactory, especially Qinghai-Tibet Plateau with relatively lower accuracy because of the unavailable observation data. Overall, the CASA-model is much more ideal for estimating NPP all over China in the absence of data. This study provides a comprehensive intercomparison of different NPP-simulated models and can provide powerful help for researchers to select the appropriate NPP evaluation model.

## Introduction

NPP is the amount of organic matter produced by photosynthesis minus autotrophic respiration, which is defined as the net amount of organic matter fixed by plants through photosynthesis. It represents the net carbon flow from the atmosphere to the terrestrial ecosystems and is affected by many factors, such as climate, soil, nutrients and $CO_2$ [1–3]. Generally, assuming that vegetation can make full use of the climate resources, such as light, heat, and water when other factors are in the optimum state, which can obtain the maximum biological or

https://lpdaac.usgs.gov/products/mod17a3hv006/#tools Then, click "access data"—Data Pool. You'll see the description: The Data Pool is the publicly available portion of the LP DAAC online holdings. Data Pool provides a direct way to access data product files via HTTPS. All Data Pool holdings are available at no cost. 2. http://www.resdc.cn/Login.aspx?ReturnUrl=http%3a%2f%2fwww.resdc.cn%2fUser%2fUserEdit.aspx The study used two pieces of data in resdc. (1)Spatial interpolation dataset of annual precipitation in China since 1980 download: https://www.resdc.cn/data.aspx?DATAID=229 (2)A spatially interpolated dataset of annual mean temperature in China since 1980 download: https://www.resdc.cn/data.aspx?DATAID=228.

**Funding:** The Scientific and Technological Innovation Team of Universities in Henan Province (No.22IRTSTHN008) provided the contributors with financial and equipment.

**Competing interests:** The authors have declared that no competing interests exist.

agricultural yield per unit area of land is called climatic potential productivity [4,5]. As one of the critical indicators of ecosystem function, NPP reflects the growth of vegetation and the health status of ecosystems [6,7], and it is useful for modeling researches of regional and global carbon cycle. Therefore, a better understanding of NPP estimates is essential for the prediction of the future carbon budgets in the context of global warming.

Traditionally, NPP estimations were based on field surveys and observations. However, these methods are not feasible on a large scale because of its low efficiency, high cost, and inability [8,9]. With the development of satellite remote sensing technology, we can find a powerful way to access NPP in a large scale. MOD17A3-NPP, as one existing large scale product acquired by remote sensing, has 1 km resolution and complete applications in different ecosystems [10–12]. It has been widely used to reflect the response of vegetation to climate change [13–16]. Generally, there are large uncertainties due to the lack of data in the large-scale NPP calculation [17–19]. Therefore, models that require fewer data have a more significant advantage, such as climate productivity models: Miami model [20], Thornthwaite Memorial model [20], synthetic model, the light use efficiency models [21] and so on. Climate productivity models are always used to estimate potential productivity as the maximum regional productivity [3]. The synthetic model is established mainly based on the measured biomass data, which is from 125 stations connected with natural mature and 23 stations related to natural vegetation NPP in China such as grassland, and desert [6]. Meanwhile, there is a notable advantage for estimating actual productivity based on the CASA model [22–25]and the current version of the CASA model take into account land-cover change [26–29]. However, there are many differences and uncertainties during these models. Systematic validation of those models is rare in large-scale regions, which may influence our understanding of the ecosystem's carbon balance and assess vegetation response to climate change. Moreover, most studies have focused on a single NPP estimated model and the results of various methods used to have considerable differences in uncertainty. Evaluating the reliability of results acquired by different models and assessing the differences are still open questions.

In this study, we used meteorological data as input data, the estimated NPP using climate productivity model (Thornthwaite Memorial model), synthetic model and CASA model were compared with MOD17A3-NPP as the reference data over China during the past 16 years (2000–2015). Then the statistics evaluation metrics (Relative Bias (RB), Pearson linear Correlation Coefficient (CC), Root Mean Square Error (RMSE), and Nash-Sutcliffe efficiency coefficient (NSE)) for three models with the reference data were calculated.

## Materials and methods

### Study area

China, the third-largest country globally, is located in the East of Asia on the western shore of the Pacific Ocean [30]. To clearly to distinguish NPP in China under other climatic and topographic conditions, we divide China into seven major regions and each of which may contain one or more administrative areas. The seven significant regions include: the Xinjiang (XJ) region, which has arid and semi-arid climate characteristics. Qinghai–Tibet Plateau (TP) which has an average elevation of about 4500m. Northwestern China (NW) bounded by the 400 mm annual precipitation isohyet. Northeastern China (NE) is located in the north of the Yan mountains. Northern China (NC) is located in the north of the Qinling Mountains–Huai River line and the vegetation patterns in this region are characterized by a mosaic of agricultural vegetation. Yunnan–Guizhou Plateau in southwestern China (SW)is bounded by the Tapa Mountains and Wulingshan Mountains to the north and east. Southern China (SC) south of the Nanling Mountains and Southeast of the Wuyi Mountains. Southern China (SC) south

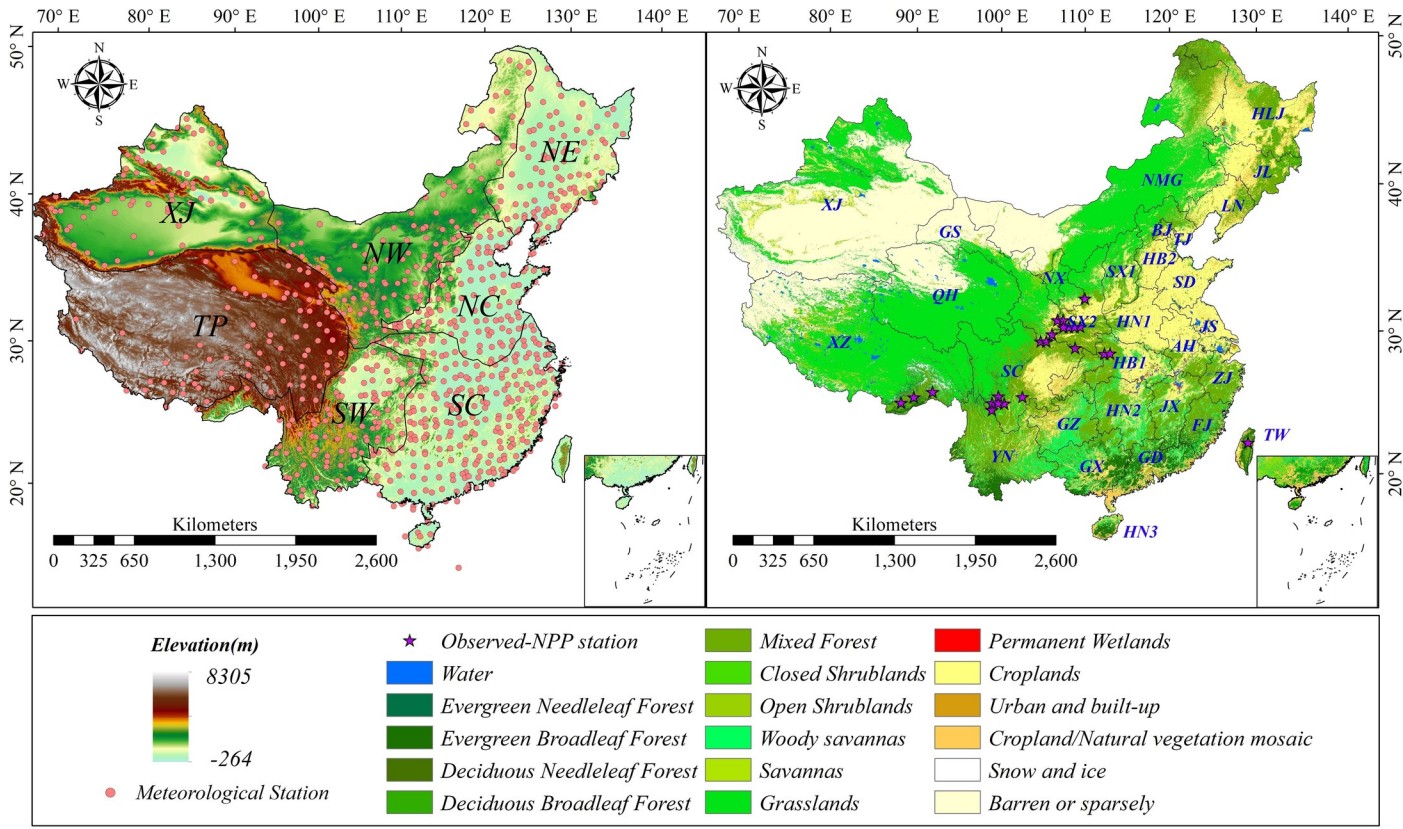

**Fig 1.** (a). Different regions of the study area according to the elevation and annual precipitation. The pink circles are 839 meteorological stations. (b). Land use of China. The 23 stars represent the observed-NPP stations from published papers, and these stations located in the temperate forest are mainly used to verify the potential productivity.

of the Nanling Mountains and Southeast of the Wuyi Mountains. The same China divisions are from [31,32].These subregions abbreviations are labeled in Fig 1A and will be used herein.

## Data and processing

**Remote sensing data.** *MODIS NPP data*. The MOD17A3's global NPP data for 2000–2015 was used as the reference data in this study. These data were obtained from NASA's website (https://lpdaac.usgs.gov/data_access/) with 1 km spatial resolution. It contains total primary productivity (GPP), net primary productivity (NPP) and net direct quality control (NP_QC). In this study, 21 images in China were selected. The NPP data from 2000 to 2015 were converted into NPP data with unit g C·m$^{-2}$·yr$^{-1}$, and the scale coefficient was 0.1(Fig 2). Studies found that MOD17A3 NPP dataset has outstanding agreements with the observations at the global or country scale [33,34].

*MODIS NDVI data*. MODIS normalized difference vegetation index (NDVI) product with a 250m/16-day spatiotemporal resolution between 2000 and 2015 was used in this study. These data were from the MODIS product MOD13A1. The monthly NDVI data was generated by the maximum-value composite method [3] and then was reprojected to the Albers equal-area projection. The NDVI data were then used to drive CASA-model as input data for NPP estimation.

**Meteorological data.** This study used meteorological data such as daily temperature, daily precipitation, and solar radiation from 2000 to 2015. All data is provided by the China

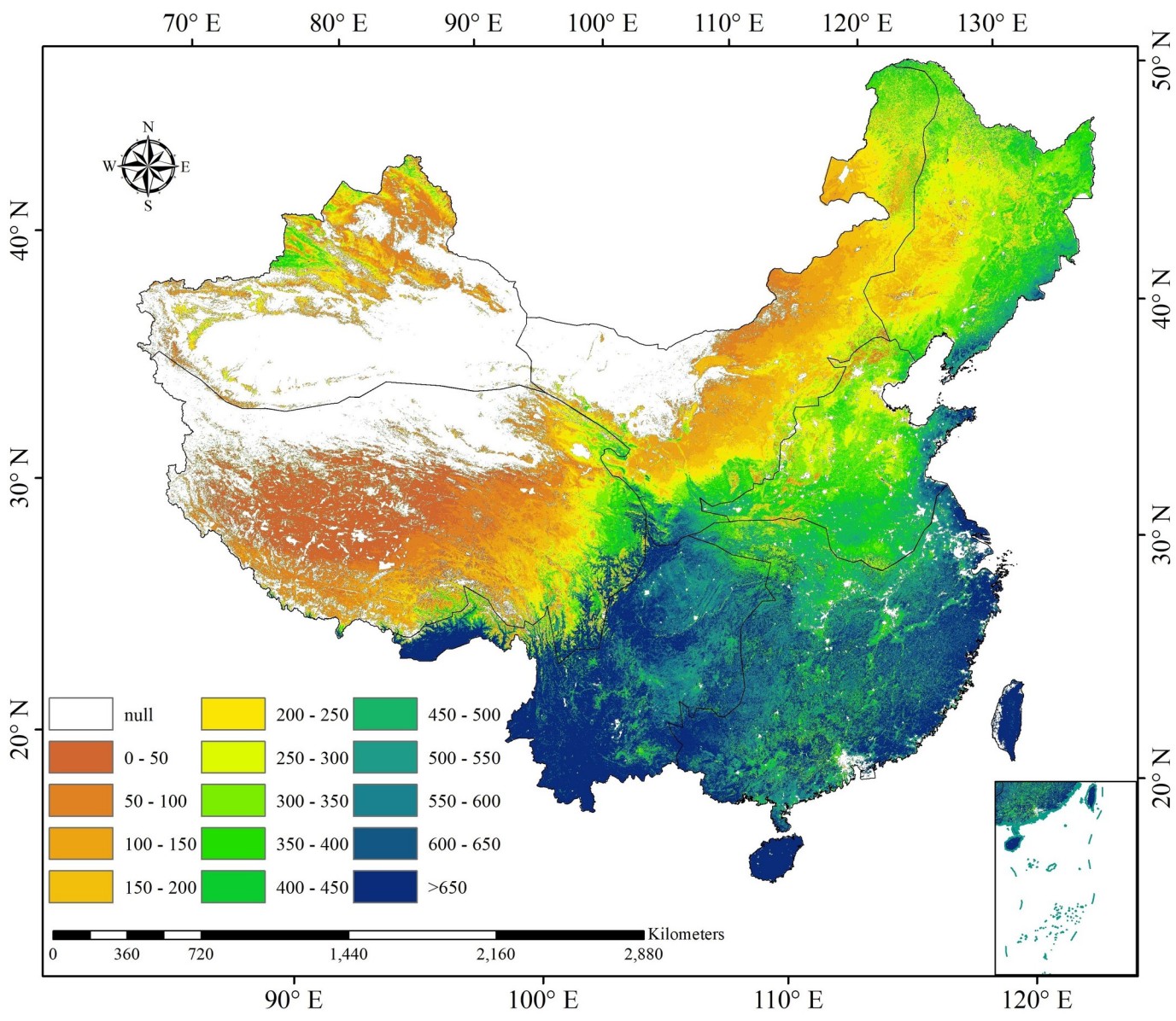

**Fig 2. The spatial distribution of multi-year MOD17A3-NPP average from 2000 to 2015.**

Meteorological Administration (http://cdc.nmic.cn/home.do), obtained from the 839 meteorological stations (130 solar radiation stations) in the whole country. Meteorological data driving the CASA model include monthly precipitation, monthly mean temperature, and monthly solar radiation. The data supplied to drive the Thornthwaite Memorial and Zhou model are only yearly temperatures and precipitation datasets. These data were interpolated using ANUSPLIN (version 4.2) to create regular monthly data or yearly data layers with the same spatial resolution as MOD17A3-NPP.

**Land use data and others.** Land use maps were from the MODIS product of MCD12Q1 and obtained by NASA (https://lpdaac.usgs.gov/data_access/) with 1km resolution. The vegetation are classified into 11 categories according to IGBP global vegetation classification scheme, including the Evergreen Needleleaf Forest, Evergreen Broadleaf Forest, Deciduous

Needleleaf Forest, Deciduous Broadleaf Forest, Mixed Forest, Closed Shrublands, Open Shrublands Woody Savannas, Savannas Grasslands, Permanent Wetlands, Croplands, Cropland/natural vegetation mosaic (Fig 1B). There are not including Water, Urban/build-up, Snow and ice, and Barren or sparsely.

**In-situ survey data.** In this study, the verification NPP data were derived from the study [35], with the Global Primary Productivity Initiative (https://daac.ornl.gov/). These data come from the National Forest Resources Inventory conducted by the Chinese Forestry Department during the period 1989–1993. Besides, we also used in-situ survey datasets from published kinds of literature with well-documented field sites [3,36–38]. These data provided some valuable information such as site names, latitude, longitude, elevation, biomass and NPP estimations for most of the plant components. Mostly, the data from Global Primary Productivity Initiative covered representative sample points of different vegetation types, and the data are widely used in global model parameterization and result verification. All data included different vegetation types and total NPP data of China's administrative districts. Finally, 23 observed-NPP stations (Fig 1B) were collected as NPP validation data.

## Methods

In this study, we used climate productivity model (Thornthwaite Memorial model), synthetic model and CASA model to calculated NPP over China during the past 16 years (2000–2015). Then the comparative analysis was employed to assess the performance of the NPP simulated model with MODIS NPP. The statistics evaluation metrics (Relative Bias (RB), Pearson linear Correlation Coefficient (CC), Root Mean Square Error (RMSE), and Nash-Sutcliffe efficiency coefficient (NSE)) were calculated. The flowchart of the the methodology employed in this study is as follow (Fig 3):

**Climate productivity model for estimating NPP.** Thornthwaite Memorial [39], as one climate productivity model, established a statistical relationship between Net Primary Productivity (NPP) and evapotranspiration (ET) based on the relationship between evapotranspiration, temperature, precipitation, and vegetation. On this basis, Lieth [40] proposed the Thornthwaite Memorial model in 1975 based on the vegetation NPP in 50 different locations on 5 continents. The climate factors considered in this model are relatively simple and can better reflect the key factors affecting plant growth and development, such as temperature, precipitation, and evapotranspiration. The calculation formula is∉

$$NPP_T = 3000 \times [-e^{-0.0009695(v-20)}] \tag{1}$$

$$v = \frac{1.05r}{\sqrt{1 + \left(1 + \frac{1.05r}{L}\right)^2}} \tag{2}$$

$$L = 300 + 25T + 0.05T^3 \tag{3}$$

where, v represents annual average actual evapotranspiration (mm), r is annual average precipitation (mm), L is the maximum annual evapotranspiration (mm), T is the average annual temperature (°C). $NPP_T$ is calculated in units of g DW/m$^2$/yr. This was implemented by applying a conversion factor of 0.475 in China [26] from dry matter (DW) to carbon content (g C·m$^{-2}$·yr$^{-1}$).

**Synthetic model for estimating NPP.** Zhou [41] and Zhang [42] based on the energy balance equation and the water balance equation established the NPP model of natural vegetation combining the physiological and ecological characteristics of plants and the relationship

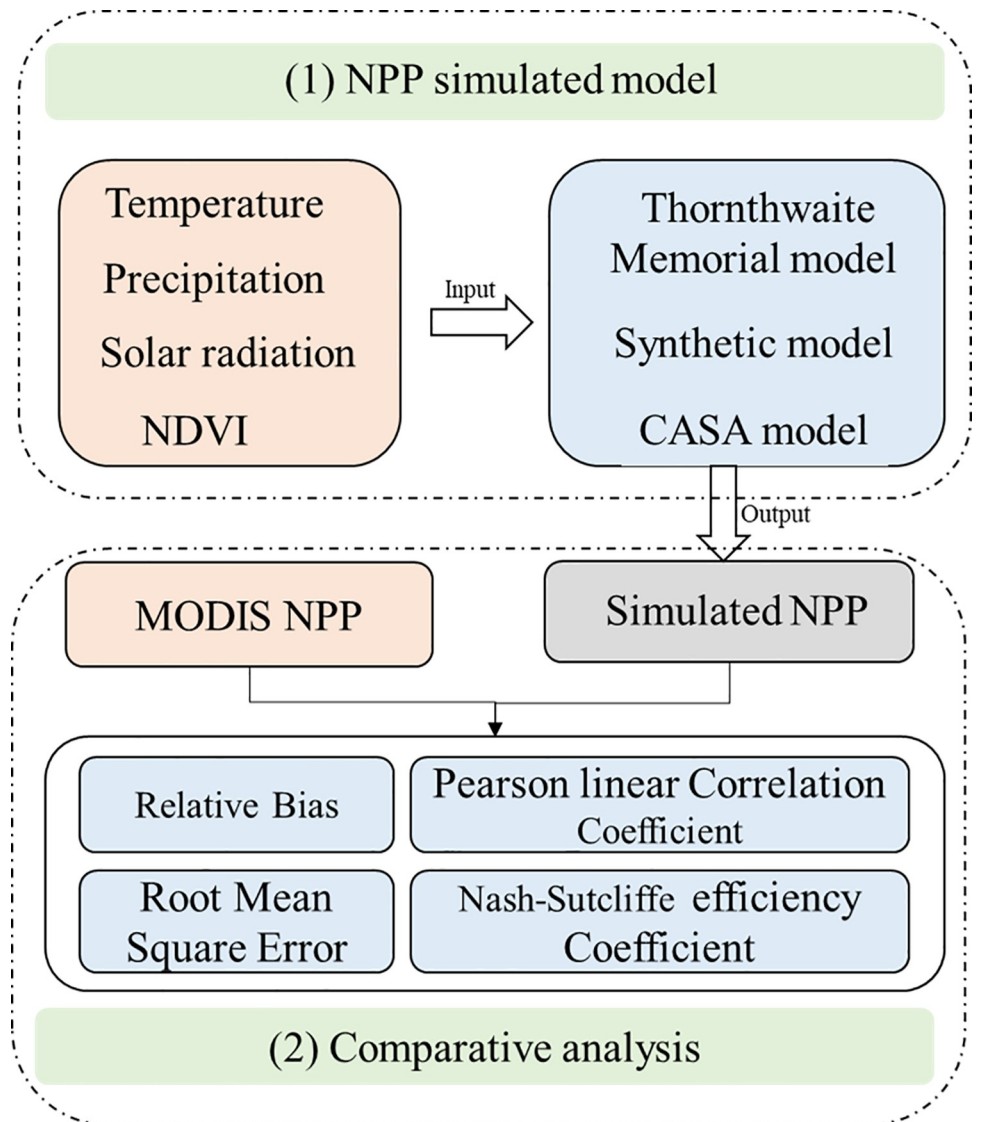

**Fig 3. Flowchart of the methodology employed in this study.** Light green represents the process of this study; Light orange represents data used in this study; Gray represents the output; Light blue represents the methods.

between water and heat balance. The calculation formula [3] is as follows:

$$NPP = 100 \times RDI \frac{rR_n(r^2 + R_n^2 + rR_n)}{(R_n + r)(R_n^2 + r^2)} exp(-\sqrt{9.87 + 6.25RDI}) \tag{4}$$

$$R_n = RDI \times r \times L \tag{5}$$

$$RDI = (0.629 + 0.237PER - 0.00313PER^2)^2 \tag{6}$$

$$PER = \frac{PET}{r} = \frac{BT \times 58.93}{r} \tag{7}$$

$$BT = \frac{\sum t_d}{365} \tag{8}$$

$$L = 597 - 0.6T_m \tag{9}$$

Where NPP was calculated in units of g DW·m$^{-2}$·yr$^{-1}$ and was implemented by applying a conversion factor of 0.475 in [26] China from dry matter (DW) to carbon content (g C·m$^{-2}$·yr$^{-1}$). $r$ is average annual precipitation, $R_n$ is annual net radiation, $RDI$ is radiant dryness, $L$ is the latent heat of evaporation, $PET$ is potential evapotranspiration, $PER$ is probably evapotranspiration rate. Biological temperature (BT) is the average temperature experienced during plant growth, generally between 0 and 30°C. Mean daily temperature ($t_d$) and mean monthly temperature ($T_m$) takes 0°C when it lowers than 0°C and can be calculated at 30°C when the temperature is higher than 30°C.

**CASA-model for estimating NPP.** The Carnegie-Ames-Stanford Approach (CASA) models [13,21], a satellite-based photosynthetic utilization models, is widely used to calculate the NPP. The CASA model requires the following parameters, such as temperature, rainfall, solar radiation, NDVI, etc. The model can be calculated by APAR (Absorbed Photosynthetic Active Radiation) times the light energy conversion rate ε. The calculation expression is as follows:

$$\text{NPP}_{(x,t)} = \text{APAR}_{(x,t)} \times \text{LUE}_{(x,t)} \tag{10}$$

$$\text{APAR}_{(x,t)} = \text{SOL}_{(x,t)} \times \text{FPAR}_{(x,t)} \times r \tag{11}$$

$$\text{FPAR}(x,t) = \frac{[\text{NDVI}(x,t) - \text{NDVI}_{i,\min}] \times (F_{\max} - F_{\min})}{\text{NDVI}_{i,\max} - \text{NDVI}_{i,\min}} + F_{\min} \tag{12}$$

$$\text{LUE}(x,t) = \text{T}_{\varepsilon 1(x,t)} \times \text{T}_{\varepsilon 2(x,t)} \times \text{W}_{(x,t)} \times \varepsilon\max \tag{13}$$

where, NPP(x,t), $APAR_{(x,t)}$ (MJ/m$^2$/month) and $LUE_{(x,t)}$ (g C/MJ) are the APAR and LUE of the vegetation in the geographic coordinate system at location x and time t. ε(x,t) represents actual the utilization of light energy. SOL(x,t) represents the total solar radiation of pixel x in month t. PAR is the incident photosynthetically active radiation (M J m$^{-2}$) per month. FPAR (x,t) is the fraction of PAR absorbed by the vegetation canopy. Constant r≈0.5 represents the solar effective radiation ratio that vegetation can utilize, namely, the ratio of PAR divided by SOL. The detail of the CASA-model can be found in the study of Potter [21] and Zhu [26].

**Validation and statistical evaluation metrics.** In this study, we validated the estimated NPP by comparing it with measured data. These data were collected from Luo's investigation data and other published literature. A series of traditional error indexes, which include Bias, Relative Bias(RB), Pearson linear Correlation Coefficient (CC), Root Mean Square Error (RMSE), and Nash-Sutcliffe efficiency coefficient (NSE), is calculated at a pixel in this study. RB, NSE, and CC are dimensionless, and RMSE is in g C/m$^2$. RB, when multiplied by 100, denotes the degree of overestimation or underestimation in percentage. The definition of RB, CC, and RMSE can be found in [31,43,44], and NSE that is generally used to verify the quality

of the hydrological model simulation results are defined as follows:

$$\text{NSE} = 1 - \frac{\sum_{t=1}^{T} \left( Q_0^t - Q_m^t \right)^2}{\sum_{t=1}^{T} \left( Q_0^t - \bar{Q}_0 \right)^2} \tag{14}$$

Where $Q_0$ represents the result of the reference model; $Q_m$ is the simulated value of the comparison model; $\bar{Q}_0$ the annual mean of the model data;t represent the time at year scale; The value range of *NSE* is (-∞, 1), and the closer the value is to 1, the higher the similarity between the comparison models and the reference models are. The closer the value is to 0, the closer the simulation result of the comparison model is to the result of the reference model, that is, the more reliable the overall result is. When the value of *NSE* is far less than 0, it indicates that the model is not credible [44].

## Results and discussion

### Validation of estimated NPP

The NPP product from MODIS has been widely used to access the response of vegetation to climate change. The validation result in this study also showed that there are good correlation between the observed data and MODIS NPP product ($R^2$ = 0.81), which reveals the NPP product from MODIS is relatively reliable. In addition, we validated the estimated NPP at the provincial scale (Fig 4A and 4B) and the station scale (Fig 4B and 4C), respectively. The results showed that the simulated NPP in each province from 2000 to 2015 using the three models in this study are highly correlated with the result of Luo during 1989 to 1993 (Fig 4B). $R^2$ was 0.52, 0.67 and 0.70, respectively. The annual average NPP from Thornthwaite Memorial model and Synthetic model are higher whereas the CASA-model are lower than those of Luo (Fig 4A) at the provincial scale. It may be caused by various uncertainties such as time inconsistency and the former two models are only taken account into climate factors, which is used to simulate the potential productivity. Fig 4C and 4D illustrated CASA model was better than the other two models at the station scale and the results simulated by CASA model were more closer MODIS NPP product, which is likely due to CASA-model and MOD17A3 product both belong to LUE model. Besides, the total NPP of China simulated by the Thornthwaite Memorial model, Synthetic model and CASA-model were 4.03Pg C (1Pg C = $10^{15}$g C), 2.54 Pg C and 3.58 Pg C, respectively. This is within the reported values of 1.95–6.13 Pg C [26,45,46]. We also compared the NPP calculated from the three models with other simulation results (Table 1), which also indicated the reliability of our results.

### Spatial distribution of NPP

Fig 5 The spatial distribution of estimated NPP showed that the trend of NPP distribution in China is higher in the Southeast and lower are the northwest (Fig 5). The spatial distribution of NPP varies from year after year due to different climatic factors, topographic factors, phenological characteristics, and vegetation types. As shown in Fig 5, there are pronounced regional differences in every model, offering a gradually decreasing trend from Southeast to northwest. In southern China, the evergreen broad-leaved forest is widely distributed and rich in resources. The annual average NPP is higher than 600 g C·m$^{-2}$·yr$^{-1}$. Rich precipitation and groundwater are more conducive to the growth of vegetation [9]. However, the average annual NPP is lower than 200 g C·m$^{-2}$·yr$^{-1}$ in the northwest due to poor soil, low temperatures and low rainfall [26]. Overall, the maximum value of NPP occurred in the southwest, southern China and Taiwan. The values between 600 and 700 g C·m$^{-2}$·yr$^{-1}$ were located in the south of the lower reaches of the Yangtze River, east of Yunnan-Guizhou Plateau and north of Nanling

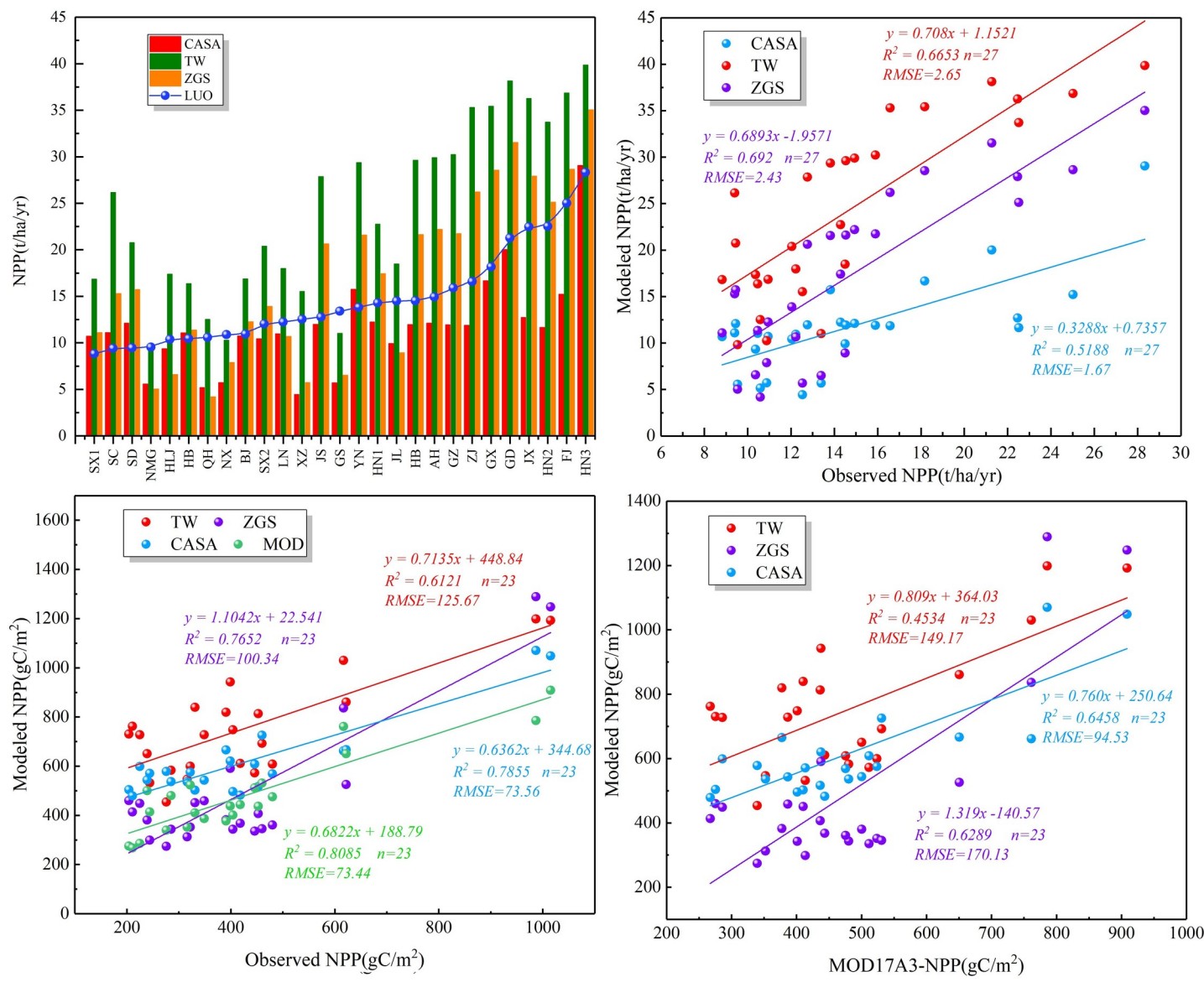

**Fig 4.** Validation of NPP (a). The range of NPP of three models and measured data in Luo's [35] study in 27 provincial administrative regions (b). The correlation of NPP between three models and Luo's measured data in 27 provincial administrative regions (c). The correlation of NPP between three models and MODIS NPP in 23 observed stations (d). The correlation of NPP between three models and MODIS NPP in 23 observed stations.

Mountains with annual NPP. Annual NPP between 400 and 600 g C·m$^{-2}$·yr$^{-1}$ were located in Daxing'an Mountains, Xiaoxing'an Mountains, east of Taihang Mountains, middle reaches of the Yangtze River Basin, most area of Sichuan, southeastern Tibet, Tianshan Mountains in Xinjiang and Altai Mountains. The low-value sites mainly distribute in Inner Mongolia, Xinjiang, Qinghai-Tibet Plateau, and parts of Shanxi, Gansu, Ningxia and Shanxi provinces with annual NPP less than 200 g C·m$^{-2}$·yr$^{-1}$.

However, there was also a discrepancy in different models. Spatial patterns of NPP over China depicted by CASA-model agree with the reference MOD17A3. NPP calculated by TW model and ZGS models showed obvious banded distribution from northwest to Southeast. Besides, the NPP contour of these two models increased steadily from northwest to Southeast,

**Table 1. Comparison of annual average NPP between this study and other studies.**

| Reference | Studied period | Studied area | model | Precision |
|:---:|:---:|:---:|:---:|:---:|
| **This study** | 2000–2015 | China | TW model<br>ZGS model<br>CASA model | $R^2 = 0.61$<br>$R^2 = 0.76$<br>$R^2 = 0.78$ |
| [39] | 2001–2010 | Heihe River Basin | TW model | None |
| [3] | 2000–2015 | The Loess Plateau | ZGS model | $R^2 = 0.734$, P <0.01 |
| [3] | 2000–2015 | The Loess Plateau | CASA model | $R^2 = 0.817$, P <0.01 |
| [46] | 2001–2010 | China | CASA model | r = 0.733,P<0.001 |
| [47] | 2000–2014 | The Ili River Valley | CASA model | $R^2 = 0.65$, P <0.01, |
| [26] | 1982–2000 | China | CASA model | RB = 4.5% |

Note: For charting and table simplification, TW represents the climate productivity (Thornthwaite Memorial) model. ZGS represents Synthetic model proposed by Guangsheng Zhou [41], the same below. The $R^2$ at station scale used in this table.

which was relatively smooth. The simulation results of the CASA model showed a larger zigzag shape, especially in the southern region. Obviously, NPP from TW model overestimated compared with the other two models, which is consistent with previous studies [24,33,36,41] As far as the input data is concerned, NPP simulated by CASA model and MODIS NPP products consider not only meteorological factors, but also different vegetation types and land surface information, so the results are more realistic. Table 2. showed the RB between NPP from three models and MODIS NPP product. Interestingly, the overall RB of ZGS model (18.54%) less than CASA-model (21.34%) and TW model (30.34%), which indicated that the ZGS model was underestimated whereas CASA-model and TW model overestimated from the country scale compared with MODIS NPP. The relative precision decreased from 81.47% in the ZGS model to 78.66% in CASA-model and 69.66% in TW model. Regionally, the results from TW model were overestimated in most regions, and the relative deviation are very high expectations for the region of SW (8.08%), which was only higher than that of CASA-model (6.86%) simulation in this region. The results of CASA-model were also largely overestimated compared with the reference data in most regions, especially in XJ, TP, NW, arid areas of Northwest China and plateau areas. The annual average estimation NPP of ZGS models on SC, NC, TP, and NW were in good agreement with the reference data, and the relative deviation in other regions were more than 30%. The results of ZGS model in regional scale performed well is likely because this model is established based on vegetation in China [39].

## Subregional statistics evaluation of NPP

NSE (Fig 6) for the simulation of three models showed the results from CASA-model was more consistent with MODIS NPP. This was likely due to the models of two results were belonged to the LUE model. However, the NSE of CASA-model was close to 0 or negative infinite values in a few areas, such as the Qinghai-Tibet Plateau, which indicated that there was inadaptable in CASA-model in this area. This is likely because the meteorological stations are few and unevenly distributed, resulting in the error of interpolation results. The maximum value of NSE from the ZGS model can reach 0.73 in the northeast and south of NE and NW. And it is greater than 0.5 in most areas of SW, the eastern part of TP and the edge of Xinjiang. Some researches indicated that the performance levels were defined as follows:
NSE > 0.65 = excellent, 0.65 ~ 0.5 = very good, 0.5 ~ 0.2 = good, and < 0.2 = poor [44,48]. NSE in most areas of central NC and some areas of southern NC was near 0.5, which indicated that ZGS model had certain applicability in these areas. However, the simulation results of

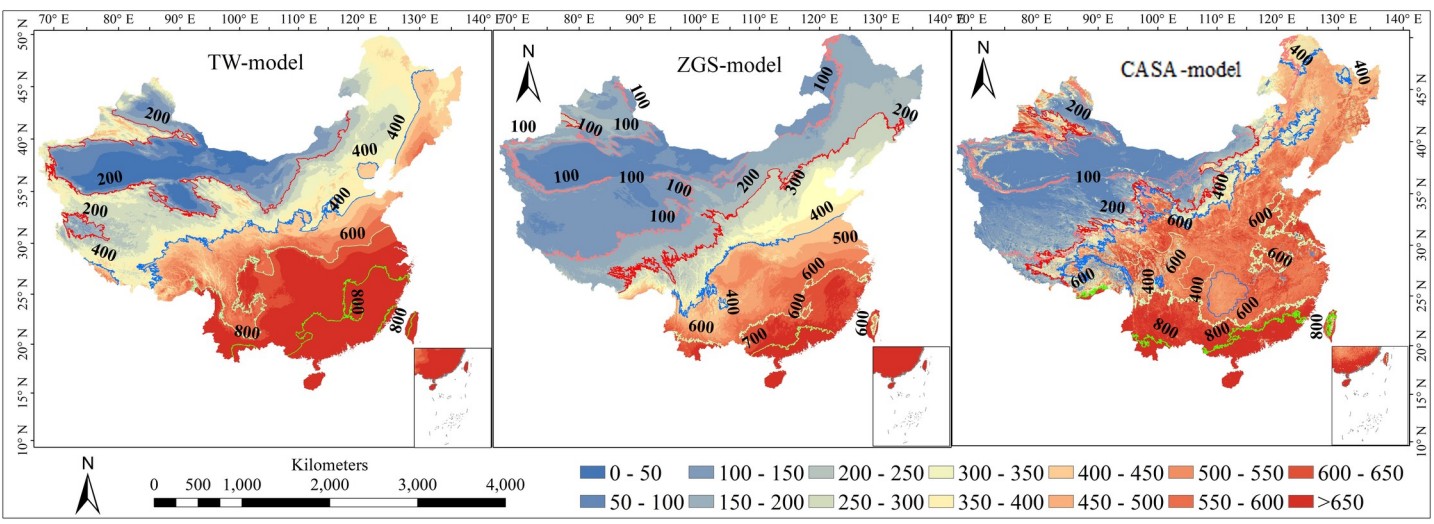

**Fig 5. The spatial pattern of NPP calculated by three models (Abbreviation: TW represents Thornthwaite Memorial model, ZGS represents the synthetic model).**

some areas such as the central and western parts of TP, the edge parts of Xinjiang and the parts of NC and SC border appear near 0. Compared with the simulation results of the two models mentioned above, the NSE simulation results of the TW model were less than 0.5, which illustrated that there were great differences between NPP calculated by TW model and MODIS NPP product.

The average RMSE (Fig 7) of the three models was less than 200 g $C·m^{-2}·yr^{-1}$, within the allowable range of their respective errors in the whole country. Table 3 indicated that CASA-model performed well noticeably in NE, XJ, SW and NW. The average of RMSE increased from 111.96 g $C·m^{-2}·yr^{-1}$ in CASA-model to 133.14 g $C·m^{-2}·yr^{-1}$ in ZGS model and 172.46 g $C·m^{-2}·yr^{-1}$ in TW model. It also showed NPP calculated by CASA-model was consistent with the reference data. In terms of spatial distribution, the RMSE of CASA-model was 0–150 g $C·m^{-2}·yr^{-1}$ in the whole region, especially in Inner Mongolia Autonomous Region, XJ, most areas of NE and Shandong Peninsula where RMSE was less than 50 g $C·m^{-2}·yr^{-1}$. RMSE was more than 200 g $C·m^{-2}·yr^{-1}$ in the central of NC and TP. The larger RMSE occurs in southern SC, Southern SW and southern Tibet. However, the ZGS model also had preponderance in most areas of NC, XJ and TP, and the range of RMSE was between 150–250 g $C·m^{-2}·yr^{-1}$ in the northeastern margin area. The differences between simulated NPP and reference data were more considerable in most areas of SC and the junction area between TP and SC where RMSE was more significant than 400 g $C·m^{-2}·yr^{-1}$. Compared with the above two results, the RMSE

**Table 2. RB for the results of three models comparable with reference data (Note: TW represents Thornthwaite Memorial model).**

|  | MODIS_NPP (g C/m²) | TW_NPP (g C/m²) | TW_RB (%) | ZGS_NPP (g C/m²) | ZGS_RB (%) | CASA-NPP (g C/m²) | CASA-RB (%) |
|---|---|---|---|---|---|---|---|
| **SC** | 617.34 | 771.00 | 24.89 | 596.07 | -3.45 | 682.98 | 10.63 |
| **NE** | 318.24 | 381.16 | 19.77 | 177.31 | -44.28 | 357.95 | 12.47 |
| **NC** | 351.80 | 470.72 | 33.80 | 353.65 | 0.53 | 461.89 | 31.29 |
| **XJ** | 58.01 | 136.84 | 135.89 | 76.52 | 31.91 | 119.71 | 106.36 |
| **TP** | 109.24 | 334.50 | 206.21 | 114.17 | 4.51 | 223.48 | 104.58 |
| **SW** | 578.08 | 531.38 | -8.08 | 338.64 | -41.42 | 538.45 | -6.86 |
| **NW** | 167.79 | 242.66 | 44.62 | 136.12 | -18.87 | 285.67 | 70.25 |
| **ALL** | 314.36 | 409.75 | 30.34 | 256.07 | 18.54 | 395.73 | 21.34 |

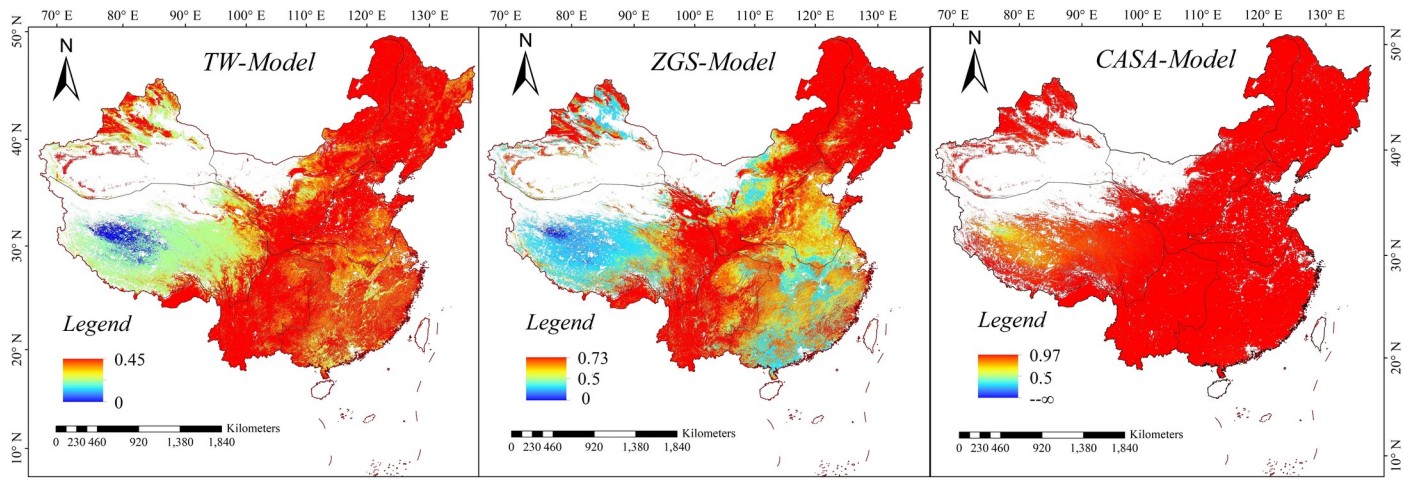

**Fig 6. The spatial distribution of NSE of three models compared with MODIS NPP product.**

of TW model was higher in the whole country. In summary, the error of CASA-model and MOD17A3 reference data is the smallest, followed by ZGS model, and the worst is TW model.

CC in the CASA-model, TW model and ZGS model were not good consistent with the reference data. The average CC of the CASA-model was 0.51 higher than that of ZGS and TW model (Fig 8) in the whole country. However, the average values showed large regional differences. The CC of three models in NW and NE regions was higher than the other regions, which revealed that estimated NPP in these regions using CASA-model, ZGS model and TW model were well consistent with MODIS NPP. The spatial distribution of CC in three models compared with MODIS NPP product (Fig 9) showed there are a good consistency in most areas of NE, NW, central NC and SW where CC can reach more than 0.8, showing a positive correlation. The overall performance of CASA-model is a positive correlation with MODIS NPP product in most part of China.

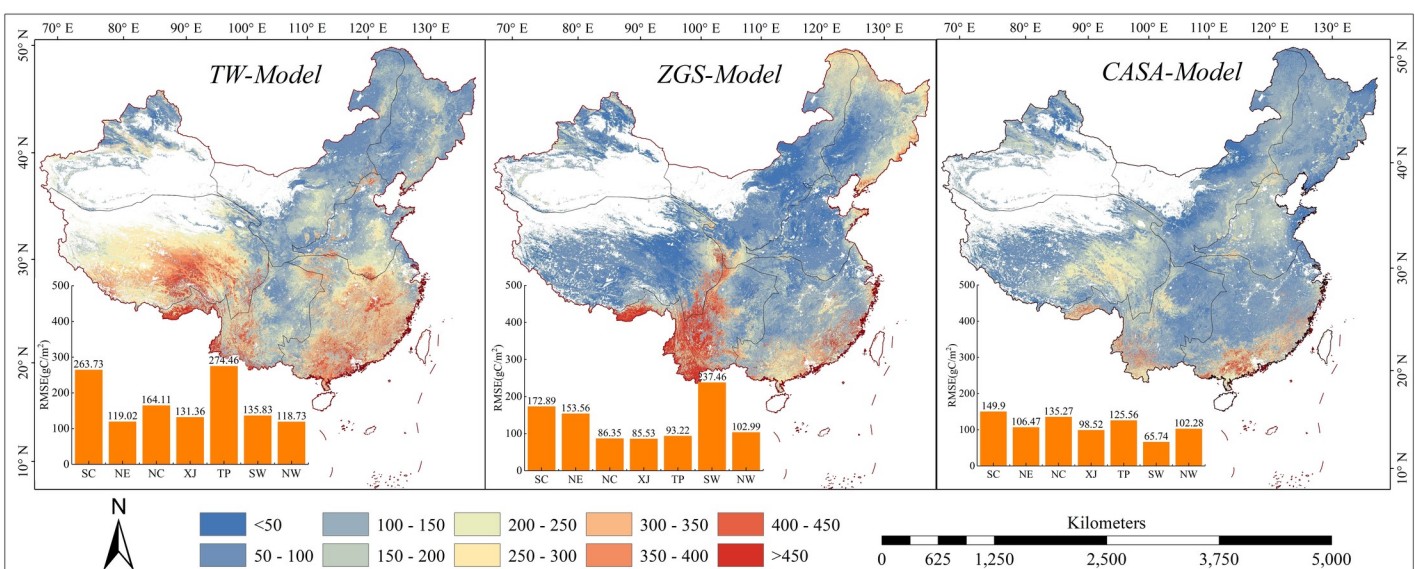

**Fig 7. The spatial distribution of RMSE of three models compared with MODIS NPP product.**

**Table 3. RMSE of three models comparable with MODIS NPP product.**

| | TW-RMSE (g C/m²) | ZGS-RMSE (g C/m²) | CASA-RMSE (g C/m²) |
|---|---|---|---|
| **SC** | 263.73 | 172.89 | 149.9 |
| **NE** | 119.02 | 153.56 | 106.47 |
| **NC** | 164.11 | 86.35 | 135.27 |
| **XJ** | 131.36 | 85.53 | 98.52 |
| **TP** | 274.46 | 93.22 | 125.56 |
| **SW** | 135.83 | 237.46 | 65.74 |
| **NW** | 118.73 | 102.99 | 102.28 |
| **Average** | 172.46 | 133.14 | 111.96 |

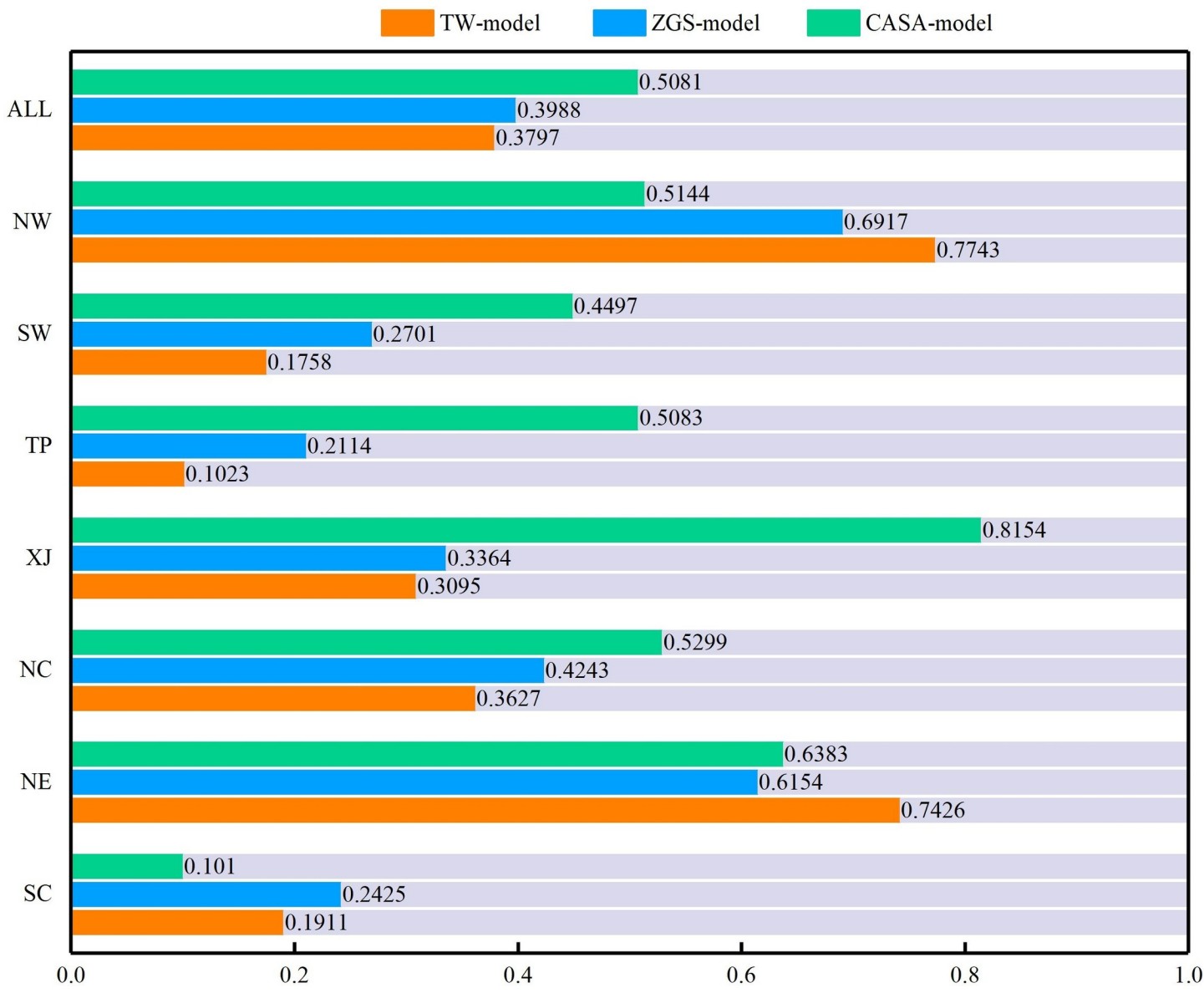

**Fig 8. CC of three models compared with MODIS NPP product.**

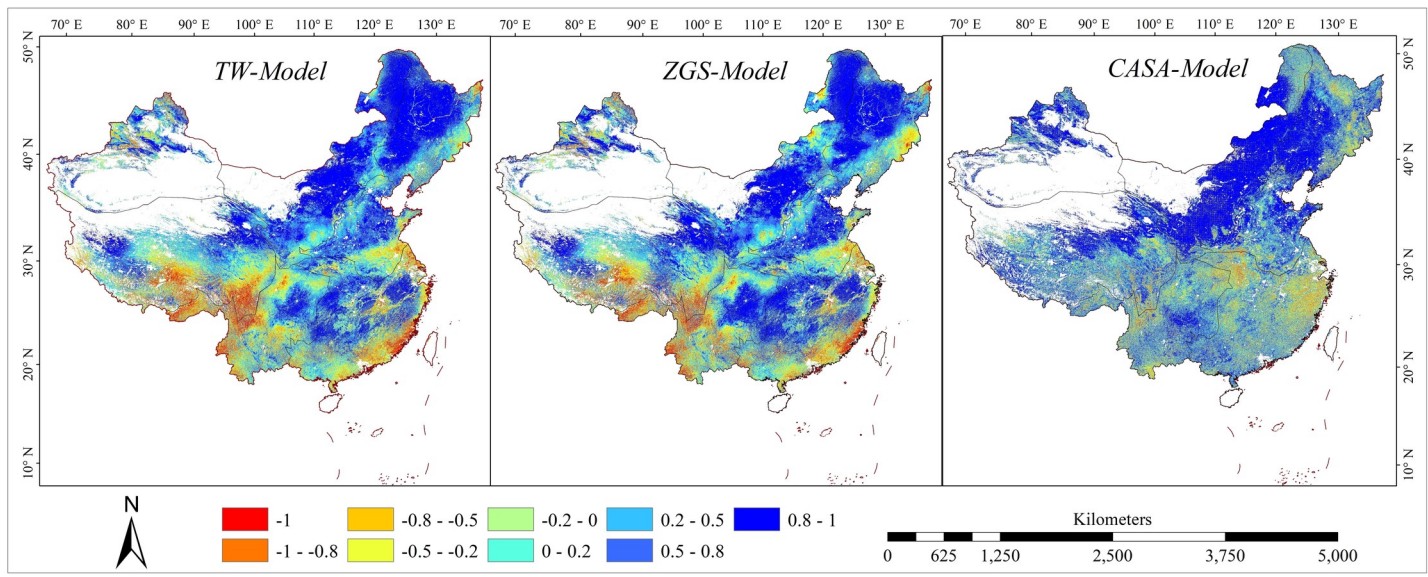

**Fig 9. The spatial distribution of CC in three models compared with MODIS NPP product.**

## Uncertainties

The formation process of vegetation NPP is affected by many factors, not only related to various physiological and ecological factors, but also related to many complex environments. However, the three models and the MODIS NPP used in this study take fewer factors into account. The three models were all relatively simple, especially the MODIS NPP was known to underestimate NPP in areas with high productivity, and overestimate NPP in low productivity areas [33].There are still large uncertainties because the real situation was not entirely the same with the simulation. Besides, many scholars have proved that there are obvious spatial-temporal variations between different vegetation types [7,26,45,49,50].Although the validation data used in this study adopt the multi-year average of eliminating systemic errors, the inconsistency of time intervals will inevitably lead to errors. Moreover, the uncertainty in simulated NPP also resulted from climate input data such as the differences in temperature, precipitation, topography and other aspects at the station scale through interpolated tools. Nonetheless, our results show that CASA model performs best among the three models for estimating NPP in the absence of parameters. This study provides new insight for large-scale and long-time series NPP evaluation and helps to understand the difference of various models and the application of models in different regions.

## Conclusions

In this study, we evaluate the effectiveness of there models (TW model, ZGS model and CASA model) compared with MODIS NPP and observed data by calculating a series of statistics evaluation metrics (RB, RMSE, NSE, CC). The multi-year average NPP from the three models over China during 2000–2015 showed that NPP simulations of the above three models were all within the reported values comparable to other's results. However, NPP calculated by CASA model performed better than TW model and ZGS model according to statistics evaluation metrics such as RB, RMSE, NSE and CC in the whole country. Meanwhile, there are regional differences in different models. The results from ZGS model and CASA-model had same advantages from a regional perspective, ZGS model had lower RMSE in the region of SC

(86.35 g C·m$^{-2}$·yr$^{-1}$), XJ (85.53 g C·m$^{-2}$·yr$^{-1}$) and TP (93.22 g C·m$^{-2}$·yr$^{-1}$) than others. In addition, the difference between the three models occurred mainly in different ecosystems. The three models revealed very high maximum at the individual pixels, especially in southeast China where there are a mixed forest, urban and built-up. In summary, the CASA-model agrees well with MODIS NPP and observed data, which can be used to estimated NPP in the absence of data. All in all, the study results will provide baseline data for large-scale and long-time series NPP evaluation and help the policymakers understand the current situation of NPP spatial distribution in China and develop environmental policies related to crop production.

## Acknowledgments

The authors gratefully acknowledge the supports by research team.

## Author Contributions

**Conceptualization:** Jinke Sun.

**Data curation:** Jinke Sun.

**Formal analysis:** Jinke Sun.

**Funding acquisition:** Haipeng Niu.

**Investigation:** Jinke Sun.

**Methodology:** Jinke Sun.

**Project administration:** Jinke Sun, Haipeng Niu.

**Resources:** Jinke Sun, Haipeng Niu.

**Software:** Jinke Sun.

**Validation:** Jinke Sun.

**Visualization:** Jinke Sun.

**Writing – original draft:** Jinke Sun.

**Writing – review & editing:** Jinke Sun, Ying Yue.

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
