## [Decision Letter · Decision Letter 0]

3 Feb 2021

PONE-D-20-31479

Spatial Pattern Change and Analysis of NPP in Terrestrial Vegetation Ecosystem based on three models in China

PLOS ONE

Dear Dr. Sun,

Thank you for submitting your manuscript to PLOS ONE. After careful consideration, we feel that it has merit but does not fully meet PLOS ONE’s publication criteria as it currently stands. Therefore, we invite you to submit a revised version of the manuscript that addresses the points raised during the review process.

We look forward to receiving your revised manuscript.

Kind regards,

Vassilis G. Aschonitis

Academic Editor

PLOS ONE

"The authors gratefully acknowledge the financial supports by the Innovative research team of Henan

Polytechnic University (Grant No. T2018-4)."

"The author received no specific funding for this work."

4. We note that Figures 1, 2, 4, 5, and 6 in your submission contain map images which may be copyrighted. All PLOS content is published under the Creative Commons Attribution License (CC BY 4.0), which means that the manuscript, images, and Supporting Information files will be freely available online, and any third party is permitted to access, download, copy, distribute, and use these materials in any way, even commercially, with proper attribution. For these reasons, we cannot publish previously copyrighted maps or satellite images created using proprietary data, such as Google software (Google Maps, Street View, and Earth). For more information, see our copyright guidelines: http://journals.plos.org/plosone/s/licenses-and-copyright.

(1) You may seek permission from the original copyright holder of Figures 1, 2, 4. 5 and 6 to publish the content specifically under the CC BY 4.0 license. 

(2)If you are unable to obtain permission from the original copyright holder to publish these figures under the CC BY 4.0 license or if the copyright holder’s requirements are incompatible with the CC BY 4.0 license, please either i) remove the figure or ii) supply a replacement figure that complies with the CC BY 4.0 license. Please check copyright information on all replacement figures and update the figure caption with source information. If applicable, please specify in the figure caption text when a figure is similar but not identical to the original image and is therefore for illustrative purposes only.

5. We noticed you have some minor occurrence of overlapping text with the following previous publications, which needs to be addressed:

- https://www.sciencedirect.com/science/article/abs/pii/S0048969718326512?via%3Dihub

- https://ieeexplore.ieee.org/document/8948039

- https://www.mdpi.com/2072-4292/10/6/860/html

- https://www.mdpi.com/2072-4292/9/10/1082

- https://www.tandfonline.com/doi/abs/10.1080/01431161.2018.1430913?journalCode=tres20

In your revision ensure you cite all your sources (including your own works), and quote or rephrase any duplicated text outside the methods section. Further consideration is dependent on these concerns being addressed.

Reviewers' comments:

Reviewer's Responses to Questions

**Comments to the Author**

1. Is the manuscript technically sound, and do the data support the conclusions?

Reviewer #1: Yes

Reviewer #2: Yes

Reviewer #3: No

2. Has the statistical analysis been performed appropriately and rigorously? 

Reviewer #1: Yes

Reviewer #2: Yes

Reviewer #3: Yes

3. Have the authors made all data underlying the findings in their manuscript fully available?

Reviewer #1: Yes

Reviewer #2: Yes

Reviewer #3: Yes

4. Is the manuscript presented in an intelligible fashion and written in standard English?

Reviewer #1: No

Reviewer #2: No

Reviewer #3: No

5. Review Comments to the Author

Reviewer #1: The current study provides a baseline for multi-temporal and large-scale NPP evaluation. It employs 3 models, climate-related and RS-based. The results were compared with reference data and evaluated with statistical evaluation metrics as well.

Studies have indicated that climate-related models (Thornthwaite Memorial model) were based on empirical regressions between climatic conditions and measured NPP. Therefore, parameters used in these models may need to be adjusted for a specific region, otherwise, overestimations in the potential NPP may occur (Sun, Q., Li, B., Zhou, C. et al. A systematic review of research studies on the estimation of net primary productivity in the Three-River Headwater Region, China. J. Geogr. Sci. 27, 161–182 (2017). ttps://doi.org/10.1007/s11442-017-1370-z). The area under investigation of this study extends on a national scale. Based on the differences in elevation, vegetation types, climatic zones that China is characterized with, do your results indicate such an approach?

The MODIS NDVI dataset used in the CASA model has been modified using the maximum-value composite method. Please provide a reference. How many images per year were initially acquired and finally used into the composition method? Was there any pre-processing followed in order to exclude the invalid values and smooth out the noise?

Please provide the spatial resolution of the land use maps. The re-classification scheme of 14 categories (in the manuscript are 15) indicate the classes that the NPP was estimated? Overall, was the final spatial resolution of the RS data similar?

The in-situ data, used for the validation, were derived from different sources. This raises the question of whether the NPP measuring method followed and the sampling criteria per field investigation is similar so as to unify the NPP measures prior to the validation process.

Finally, there are some grammar errors in English, so a grammar check is necessary.

Reviewer #2: Please carefully review the article for typographical / grammatical and spelling issues. These issues are too numerous to include here and detract significantly from the readability of the manuscript.

In reading the manuscript I do not have any issues with the assessment as undertaken. Authors go into great detail describing the phenomenon which are clearly visible in the images and tables. I would encourage authors to guide readers in practical applications of results. In the final sentence authors state that "Importantly, these results can provide powerful help for researchers to select the appropriate NPP model evaluation." I agree this is important, but do not feel that authors have adequately armed readers with this capability in their conclusions. I encourage authors to take their statistical results a step further, and provide recommendations in a real-world practical context of how they should guide decisions on which models to use. Under what circumstances are some models better than others? This should be concluded from statistical results and made clear to readers.

Reviewer #3: This is an important study evaluating net primary productivity (NPP) based on different models. Great efforts by the authors! However, the paper still needs some revisions. While considering the following few observations and suggestions, refer to the attachment for more:

Abstract:

The Abstract lacks coherence - a general observation throughout the paper. It should be rewritten to clearly and briefly reflect the background of the study, the aim of the study, methods employed/data, synopsis of results and perhaps, conclusion, either as deduction or implication.

Methods:

While quotations might not be bad in methods, please clearly (and briefly) state steps and how each process was carried out.

Results:

It is observed that Results and Discussion are presented concurrently. Great effort here. Please clearly described your results and craft main arguments arising from the FIGURES (Results), highlighting how your findings provide the ultimate missing piece to the puzzle – research question (if any) and the knowledge gaps you may have identified.

References:

Kindly adhere to PLOS format in your in-text citation and referencing.

6. PLOS authors have the option to publish the peer review history of their article (what does this mean?). If published, this will include your full peer review and any attached files.

Reviewer #1: No

Reviewer #2: No

Reviewer #3: No

---

## [Author Response · Author response to Decision Letter 0]

1 Apr 2021

Respond to specific reviewer and editor comments":

1.The format of the paper has been modified.

2.The fund information was removed from the article.

3.Title modified: Evaluation of NPP using three models compared with MODIS-NPP data over China

4.A note on the copyright of the picture: The data used in this manuscript is publicly available , all the images were created by the author himself through experiments, don't need authorization.

5.Repeated areas of the article have been revised or cited.

Reviewer #1: The current study provides a baseline for multi-temporal and large-scale NPP evaluation. It employs 3 models, climate-related and RS-based. The results were compared with reference data and evaluated with statistical evaluation metrics as well.

Studies have indicated that climate-related models (Thornthwaite Memorial model) were based on empirical regressions between climatic conditions and measured NPP. Therefore, parameters used in these models may need to be adjusted for a specific region, otherwise, overestimations in the potential NPP may occur (Sun, Q., Li, B., Zhou, C. et al. A systematic review of research studies on the estimation of net primary productivity in the Three-River Headwater Region, China. J. Geogr. Sci. 27, 161–182 (2017). ttps://doi.org/10.1007/s11442-017-1370-z). The area under investigation of this study extends on a national scale. Based on the differences in elevation, vegetation types, climatic zones that China is characterized with, do your results indicate such an approach?

Response: Thank you very much! We think you give us a good suggestion. Lieth (1975) [1] proposed the Thornthwaite Memorial model in 1974 based on the vegetation NPP in 50 different locations on 5 continents. In this study, we focus on discuss the model performance in the absence of the data. The models used in this study just take climate factors and NDVI into account and the climate factors considered in this model are relatively simple and can better reflect the key factors affecting plant growth and development, such as temperature, precipitation, and evapotranspiration. Meanwhile, the models also used in many researches [1-3], and the results in our studies showed the NPP calculated climate-based model were 4.03Pg C (1Pg C=1015g C), 2.54 Pg C, which is within the reported values of 1.95-6.13 Pg C [4,5]. Besides, our results also conclused that NPP calculated by Thornthwaite Memorial model performed worse than CASA model in most part of China, and we should chose the CASA model in the adbsence of the data. Even if, we also think you give us a good suggestion, we added the discusstion in the part of the uncertainties in the revised manuscript.

[1] Lieth, H., & Whittaker, R. H.. (1975). Primary productivity of the biosphere. Springer-Verlag.

[2] Han, X. M., & Yan, J. P.. (2013). Temporal and spatial response of crop climate productivity to climate changes in northeastern china. Acta Agriculturae Jiangxi. 

[3] GaoJing, & Wang, L.. (2010). A GIS based simulation on the potential climate productivity a case study in Gansu Province. IEEE.

[4] Feng, X., Liu, G., Chen, J. M., Chen, M., Liu, J., & Ju, W. M. , et al. (2007). Net primary productivity of china's terrestrial ecosystems from a process model driven by remote sensing. Journal of Environmental Management, 85(3), 563-573.

[5] F Pei., Xia, L., Liu, X., & Lao, C.. (2013). Assessing the impacts of droughts on net primary productivity in china. Journal of Environmental Management, 114(15), 362-371.

The MODIS NDVI dataset used in the CASA model has been modified using the maximum-value composite method. Please provide a reference. How many images per year were initially acquired and finally used into the composition method? Was there any pre-processing followed in order to exclude the invalid values and smooth out the noise?

Response: Thank you very much! We think you give us a good suggestion. MODIS normalized difference vegetation indexes (NDVI) product with a 250m/16-day spatiotemporal resolution. Therefore, there are 23 images per year used into the composition method. We added the detailed introduction of NDVI data in the revised manuscript. 

Please provide the spatial resolution of the land use maps. The re-classification scheme of 14 categories (in the manuscript are 15) indicate the classes that the NPP was estimated? Overall, was the final spatial resolution of the RS data similar?

Response: Thank you very much! We think you give us a good suggestion. Land use maps were from the MODIS product of MCD12Q2 and obtained by NASA (https://lpdaac.usgs.gov/data_access/) with 1km resolution. In this study, we focus on all the vegetation types, therefore, we revised the expression in the manuscript. Meanwhile, all the input data with 1 km resolution can ensure the final spatial resolution with similar resolution. In our study, NPP calculated by three models and MODIS NPP are all 1 km resolution.

The in-situ data, used for the validation, were derived from different sources. This raises the question of whether the NPP measuring method followed and the sampling criteria per field investigation is similar so as to unify the NPP measures prior to the validation process.

Response: Thank you very much! We think you give us a good suggestion. In this study, most of the observed data are from Luo’s study and the National Forest Resources Inventory conducted by the Chinese Forestry Department during the period 1989-1993. Therefore, they have the same criteria to measure actual NPP. Besides, the observed NPP are from publised literature. In this study, these observed data are used to verify the simulated NPP calculated by three models. They are only within the range of NPP. Therefore, the observed data used in this study is reasonable. However, in order to increase the rigor of the article, we added some description in the part of uncertainties. 

Finally, there are some grammar errors in English, so a grammar check is necessary.

Response: Thank you very much! We think you give us a good suggestion. We revised the English writing in the manuscript.

Reviewer #2: Please carefully review the article for typographical / grammatical and spelling issues. These issues are too numerous to include here and detract significantly from the readability of the manuscript.

In reading the manuscript I do not have any issues with the assessment as undertaken. Authors go into great detail describing the phenomenon which are clearly visible in the images and tables. I would encourage authors to guide readers in practical applications of results. In the final sentence authors state that "Importantly, these results can provide powerful help for researchers to select the appropriate NPP model evaluation." I agree this is important, but do not feel that authors have adequately armed readers with this capability in their conclusions. I encourage authors to take their statistical results a step further, and provide recommendations in a real-world practical context of how they should guide decisions on which models to use. Under what circumstances are some models better than others? This should be concluded from statistical results and made clear to readers.

Response: Thank you very much! We think you give us a good suggestion. We revised our expression in the part of the results and discussion.

Reviewer #3: This is an important study evaluating net primary productivity (NPP) based on different models. Great efforts by the authors! However, the paper still needs some revisions. While considering the following few observations and suggestions, refer to the attachment for more:

Abstract:

The Abstract lacks coherence - a general observation throughout the paper. It should be rewritten to clearly and briefly reflect the background of the study, the aim of the study, methods employed/data, synopsis of results and perhaps, conclusion, either as deduction or implication.

Response: Thank you very much! We think you give us a good suggestion. We have revised the abstract in the manuscript. 

Methods:

While quotations might not be bad in methods, please clearly (and briefly) state steps and how each process was carried out.

Response: Thank you very much! We think you give us a good suggestion. We added the briefly steps of CASA model in the revised manuscript.

Results:

It is observed that Results and Discussion are presented concurrently. Great effort here. Please clearly described your results and craft main arguments arising from the FIGURES (Results), highlighting how your findings provide the ultimate missing piece to the puzzle – research question (if any) and the knowledge gaps you may have identified.

Response: Thank you very much! We think you give us a good suggestion. We revised our expression in the part of the results and discussion.

References:

Kindly adhere to PLOS format in your in-text citation and referencing. 

Response: Thank you very much! We think you give us a good suggestion. We have revised the format of reference in the manuscript.

---

## [Decision Letter · Decision Letter 1]

28 Apr 2021

PONE-D-20-31479R1

Evaluation of NPP using three models compared with MODIS-NPP data over China

PLOS ONE

Dear Dr. Sun,

Thank you for submitting your manuscript to PLOS ONE. After careful consideration, we feel that it has merit but does not fully meet PLOS ONE’s publication criteria as it currently stands. Therefore, we invite you to submit a revised version of the manuscript that addresses the points raised during the review process.

We look forward to receiving your revised manuscript.

Kind regards,

Vassilis G. Aschonitis

Academic Editor

PLOS ONE

Journal Requirements:

Reviewers' comments:

Reviewer's Responses to Questions

**Comments to the Author**

1. If the authors have adequately addressed your comments raised in a previous round of review and you feel that this manuscript is now acceptable for publication, you may indicate that here to bypass the “Comments to the Author” section, enter your conflict of interest statement in the “Confidential to Editor” section, and submit your "Accept" recommendation.

Reviewer #1: All comments have been addressed

Reviewer #3: All comments have been addressed

2. Is the manuscript technically sound, and do the data support the conclusions?

Reviewer #1: Yes

Reviewer #3: Yes

3. Has the statistical analysis been performed appropriately and rigorously? 

Reviewer #1: Yes

Reviewer #3: N/A

4. Have the authors made all data underlying the findings in their manuscript fully available?

Reviewer #1: Yes

Reviewer #3: Yes

5. Is the manuscript presented in an intelligible fashion and written in standard English?

Reviewer #1: Yes

Reviewer #3: Yes

6. Review Comments to the Author

Reviewer #1: (No Response)

Reviewer #3: There is great improvement. However, units are still written with dots as superscript, g C·m-2·yr-1. Kindly write all units appropriately.

Also, in introduction, CO2.[1-3] should be CO2[1-3]. Is there any need for the ellipsis, 16]....?

7. PLOS authors have the option to publish the peer review history of their article (what does this mean?). If published, this will include your full peer review and any attached files.

Reviewer #1: No

Reviewer #3: No

---

## [Author Response · Author response to Decision Letter 1]

7 May 2021

Response: Thank you very much! I have checked the unit(g C·m-2·yr-1) again , the format is correct, and correcting the above two errors.

---

## [Editor Report · Decision Letter 2]

11 May 2021

Evaluation of NPP using three models compared with MODIS-NPP data over China

PONE-D-20-31479R2

Dear Dr. Sun,

We’re pleased to inform you that your manuscript has been judged scientifically suitable for publication and will be formally accepted for publication once it meets all outstanding technical requirements.

Kind regards,

Vassilis G. Aschonitis

Academic Editor

PLOS ONE
---

## [Editor Report · Acceptance letter]

9 Nov 2021

PONE-D-20-31479R2 

Evaluation of NPP using three models compared with MODIS-NPP data over China 

Dear Dr. Sun:

I'm pleased to inform you that your manuscript has been deemed suitable for publication in PLOS ONE. Congratulations! Your manuscript is now with our production department. 

Kind regards, 

on behalf of

Dr. Vassilis G. Aschonitis 

Academic Editor

PLOS ONE